# Effect of Extraction Methods and In Vitro Bio-Accessibility of Microencapsulated Lemon Extract

**DOI:** 10.3390/molecules27134166

**Published:** 2022-06-29

**Authors:** Claudia Giovagnoli-Vicuña, Vilbett Briones-Labarca, María Soledad Romero, Ady Giordano, Sebastián Pizarro

**Affiliations:** 1Departamento de Ingeniería en Alimentos, Universidad de La Serena, Av. Raúl, Bitrán Nachary 1305, La Serena 1720236, Chile; vbriones@userena.cl; 2Departamento de Química Inorgánica, Facultad de Química y de Farmacia, Pontificia Universidad Católica de Chile, Santiago 7820436, Chile; 3Laboratorio de Microscopía Electrónica, Universidad Católica del Norte, Larrondo, Coquimbo 1281, Chile; msromero@ucn.cl (M.S.R.); agiordano@uc.cl (A.G.); 4Departamento de Química, Facultad de Ciencias, Universidad de La Serena, Casilla 599, Benavente 980, La Serena 1720236, Chile; sgpizarro@userena.cl

**Keywords:** citrus, microcapsule, in vitro digestion, emerging technologies, ultrasound, high hydrostatic pressure

## Abstract

The extraction of bioactive compounds from fruits, such as lemon, has gained relevance because these compounds have beneficial properties for health, such as antioxidant and anticancer properties; however, the extraction method can significantly affect these properties. High hydrostatic pressure and ultrasound, as emerging extraction methods, constitute an alternative to conventional extraction, improving extractability and obtaining extracts rich in bioactive compounds. Therefore, lemon extracts (LEs) were obtained by conventional (orbital shaking), ultrasound-assisted, and high-hydrostatic-pressure extraction. Extracts were then microencapsulated with maltodextrin at 10% (M10), 20% (M20), and 30% (M30). The impact of microencapsulation on LEs physicochemical properties, phenolics (TPC), flavonoids (TFC) and relative bio-accessibility (RB) was evaluated. M30 promoted a higher microencapsulation efficiency for TPC and TFC, and a longer time required for microcapsules to dissolve in water, as moisture content, water activity and hygroscopicity decreased. The RBs of TPC and TFC were higher in microcapsules with M30, and lower when conventional extraction was used. The data suggest that microencapsulated LE is promising as it protects the bioactivity of phenolic compounds. In addition, this freeze-dried product can be utilized as a functional ingredient for food or supplement formulations.

## 1. Introduction

In 2021, the Food and Agriculture Organization’s corporate statistics database re-ported that the world production of lemons (*Citrus limon* L.) and limes (*Citrus aurantifolia* L.) was approximately 21 million tons [1]. Chile is among the ten citrus producers in South America, exporting 366,378 tons of citrus in 2020, of which 26% was lemons [1,2]. The main varieties of lemon grown in the Chile are Genova, Eureka, Fino and Messina [3].

Lemon (*Citrus limon* L.) is the third most important species of citrus grown in the world, characterized as rich in bioactive compounds such as vitamins, dietary fiber, minerals, carotenoids, phenolic compounds and essential oils that promote health [4]. Among them, the phenolic compounds of citrus fruits have been extensively investigated, due to their bioactivities such as antioxidant activity, anti-inflammatory, antimicrobial and anti-cancer and cardiovascular disease [5]. However, phenolic compounds are sensitive to oxidation which can occur from exposure to light, oxygen, humidity, among others, based on the existence of unsaturated bonds in the molecular structures [6].

The extraction method can significantly affect the bioactive compounds and their properties. High-hydrostatic-pressure (HHPE) and ultrasound-assisted (UAE) extractions have been used to maximize the recovery of bioactive compounds present in the fruit and their by-products [7]. Both extraction techniques seek to increase the solubility of the bioactive compounds while improving mass transfer [8]. Moreover, UAE and HHPE compared with conventional extraction (heat or/and agitation) have been reported to provide higher extraction efficiency in less time, with less solvent and without high temperatures. HHPE and UAE have been successfully applied in the extraction of pectin and polyphenols from tomato peel waste [7], lycopene and flavonoids from tomato pulp [9], polyphenols from lemon, lime, tangerine and sweet orange peel [10,11], polyphenols from sweet orange [12] and anthocyanins from jambolan fruit [13].

Microencapsulation is a technique to package materials (solids, liquids or gases) in microparticles or microcapsules, to protect the core from unfavorable environmental conditions and control the material’s release [14]. Among the various techniques used for microencapsulation, freeze-drying is especially useful for preserving heat-sensitive compounds [15]. This technique consists of a multi-stage process (freezing, sublimation, desorption and storage) to stabilize the materials [16]. The structure of the wall material and its composition are the main factors that affect the efficiency of protection and controlled release. Maltodextrin has been widely used as a wall material in the microencapsulation by freeze-drying of several food ingredients, seeing as it is both an abundant and economic option [14,15]. This wall material is used due to its solubility in water, low viscosity and low sugar content [17]. Therefore, the microencapsulation of bioactive compounds is a good route to improve their stability during gastrointestinal digestion.

Bio-accessibility is the amount of a food constituent released from a food matrix in the lumen of the gastrointestinal tract, and therefore, is potentially available for absorption [18]. Natural bioactive compounds such as polyphenols are released from the fruit matrix in the small intestine by digestive enzymes and in the large intestine by bacterial microflora, making them bio-accessible and potentially bioavailable [19]. The bio-accessibility of each antioxidant differs greatly, and the most abundant antioxidants in ingested fruit are not necessarily those leading to the highest concentrations of active metabolites in target tissues, since bio-accessibility may be affected by the presence of other food components including other compounds [20].

Therefore, the main interest of this research was to study in vitro bio-accessibility of the microencapsulated phenolic and flavonoid compounds of lemon extract obtained by conventional extraction (CE), ultrasound-assisted extraction (UAE), and high-hydrostatic-pressure extraction (HHPE).

## 2. Results and Discussion

### 2.1. Effects of Extraction Method on TPC, TFC, and Post-In Vitro Digestion of Lemon Extract (LE)

Before microencapsulation, the effect of the extraction method (CE, UAE and HHPE) on total phenolic and flavonoid content of lemon extract was investigated. Based on analysis, the extraction method is an important step for the release and recovery of bioactive compounds such as polyphenols. Figure 1 shows that the extractability of the bioactive compounds had a significant increase when extracted with UAE and HHPE, from 16.8 to 29.1% for TPC and from 2.5 to 5.4% for TFC, respectively, when compared with CE.

The extractability of the bioactive compounds increased with UAE due to the cavitation process during sonication. This process causes swelling of cells or rupture of cell walls, allowing high rates of diffusion through the cell wall in the first case, or a simple washing out of cell content in the second [21]. On the other hand, HHPE increases are associated with the ability to transfer pressure uniformly and instantly to all material (Pascal’s theory). Therefore, the pressure transfer rate is fast and without stress gradients, which makes the extraction process easy and efficient [22].

The effect of an “in vitro” *simulation of gastrointestinal (GI) digestion* on TPC and TFC measurements of lemon extract is reported in Figure 1. The TPC and TFC levels decreased significantly (*p* < 0.05) after in vitro digestion. Only 55.1%, 61.5% and 61.6% of TPC were recovered (bio-accessible) from the lemon extracts obtained by CE, UAE and HHPE, respectively. Likewise, we observed significant losses in TFC after in vitro digestion in relation to their initial content of lemon extract.

The bio-accessibility of phenolic compounds can be affected due to interactions with other components in the food matrix, their polarity, molecular weight, and the glucose groups attached to the molecule. In addition, these compounds can change their structure during gastrointestinal digestion [23]. The bio-accessibility of TFC has been associated with its chemical properties because flavonoids are not very stable and interact in the food matrix with other compounds [24]. Ovando-Martínez et al. [23] reported that flavonoids present a release percentage of around 15% during gastric and intestinal phases. Aschoff et al. [25] indicated that the limiting factor in flavonoids is their low solubility in digestive fluids; therefore, the low content of flavonoids in the food matrix could be less important with respect to intestinal absorption. In general terms, the results suggest that gastrointestinal digestion affects the composition of the LE.

### 2.2. Microcapsule Analysis

#### 2.2.1. Moisture Content and Water Activity (a_w_)

The moisture content in a dry food is generally associated with its stability, quality, and composition, which could affect its storage, packaging, and processing. Therefore, the residual moisture in a dry product must be less than 6% [26]. In this research, a residual moisture content in all lemon extract microcapsules of less than 4% was obtained (Table 1). However, there were no significant differences (*p* < 0.05) among samples. These results were consistent with those obtained by other researchers [27,28].

Water activity corresponds to the amount of free water in a food system, this available water is a factor that contributes to food deterioration because it can be used by unwanted microorganisms [29]. A water activity value of less than 0.3 is the recommended limit to ensure microcapsule stability because a low a_w_ represents less free water available for microorganism growth and therefore a longer shelf life [30,31]. The microcapsules obtained with maltodextrin at different concentrations presented low values of a_w_ that varied from 0.282 to 0.299 and there were no significant differences (*p* < 0.05) among them (Table 1). The tendency of the a_w_ values obtained in this work is comparable with those observed for the microencapsulated sumac extract obtained by freeze-drying with maltodextrin at 20%, 25% and 30%, which had a_w_ values of 0.41, 0.37 and 0.16, respectively [32].

#### 2.2.2. Solubility in Water

Solubility is an relevant property to assess the microcapsule’s behavior in the aqueous phase as food microcapsules need to be highly soluble to be useful and functional [31]. In addition, high porosity in freeze-dried products plays an important role in their reconstitution, because the porous structure promotes water uptake into the matrix [33].

The water solubility of these lemon extract microcapsules shows that increasing the maltodextrin content increases dissolution time, ranging on average from 136 to 235 s for M10 and M30, respectively (Table 1). No significant differences (*p* > 0.05) were observed between M20 and M30. This result shows that the addition of a smaller amount of wall material (M20) can reduce the raw material cost without changing the solubility of the microcapsules.

The solubility was not affected by the extraction method at the same concentration of maltodextrin. Maltodextrin is a polysaccharide that is produced from vegetable starch by partial hydrolysis; its water dissolution time is greater than the natural water-soluble compounds of food matrices or their extracts that are principally composed of simple sugars [32]. Therefore, the solubility in water of microcapsules depends on the structure of the microencapsulating agent. Nunes et al. [31] observed that the addition of higher concentrations of maltodextrin to *Ilex paraguariensis* extract prior to freeze-drying increased the dissolution time of the microcapsules. On the other hand, an opposite effect was reported in freeze-dried sumac extract microcapsules [32].

#### 2.2.3. Hygroscopicity

Hygroscopicity refers to the ability of a dry product to absorb moisture from the surrounding atmosphere [34]. If this property is controlled, unwanted properties such as stickiness and caking can be avoided in the dry product [35].

The values of hygroscopicity can be seen in Table 1. Addition of maltodextrin at different concentrations (M10, M20 and M30) did not affect (*p* < 0.05) the hygroscopicity of the lemon extract microcapsules. The measured hygroscopicity varied from 6.0 to 6.5 g 100 g^−1^. Tonon et al. [34] indicated that the addition of maltodextrin reduces the hygroscopicity in dry products, due to its own low hygroscopicity. The high content of soluble solids in most fruits or their juices determines the hygroscopic character attributed to the amorphous state of the microcapsule obtained by freeze-drying, therefore influencing the characteristics of the material, such as the tendency to form agglomerates and the resulting caking phenomenon [36].

#### 2.2.4. Surface Morphology Study by Scanning Electron Microscope (SEM)

Figure 2 shows micrographs by SEM of microcapsules from lemon extract (CE, UAE and HHPE) produced with maltodextrin at 10% (M10), 20% (M20) and 30% (M30). It was observed that the external morphology of the microcapsules after freeze-drying acquired a structure of broken glass with variable dimensions. This morphology is a common characteristic of microcapsules obtained by freeze-drying [15,28,35,37,38]. The glassy structure, with an irregular shape, could protect the bioactive compounds [35].

The microcapsules showed different sizes ranging from 32.17–48.29 µm (Table 1), where the highest maltodextrin concentration presented larger microcapsules. However, a great variability in the microcapsules size was observed, associated with the grinding process after freeze-drying [39]. Chen, Chi and Xu [40] indicated that the different particle size and irregular shape may be due to the low temperature during freeze-drying and the lack of forces to break up the frozen liquid into droplets or to substantially alter their surface topology in the evaporation process.

### 2.3. Microencapsulation Efficiency (ME)

The LEs obtained by UAE and HHPE presented a significant increase (*p* < 0.05) in TPC and TFC, therefore, higher microencapsulation efficiencies were observed compared with the extract obtained by CE. This increase in the TPC and TFC levels obtained by UAE is due to the propagation of pressure waves through the solvent by cavitation, a phenomenon that causes a reduction in particle size, a breakdown of cell walls and an improvement in the mass transfer, allowing a greater release of bioactive compounds [41]. On the other hand, HHPE increases the pressure difference between the inside and outside of cells, which alters plant tissues, cell membranes and organelles, allowing the solvent to penetrate the cell and extract bioactive compounds [42].

The microencapsulation efficiency (% ME) for TPC and TFC varied according to the maltodextrin concentration used. In relation to TPC, M30 had a higher ME (93.2%) compared with M20 (89.7%) and M10 (78.9%). Regarding TFC, it was observed that M30 (83.1%) had a higher ME followed by M20 (80.4%) and M10 (72.8%). This work shows that maltodextrin is a good microencapsulation agent for LE because it presents a high protector effect on TPC and TFC when compared with other studies. Yamashita et al. [35] reported significantly lower values of ME of 76% and 68% by freeze-drying with maltodextrin at 10 or 20 DE, respectively, for phenolic compounds of a blackberry by-product. Likewise, lower efficiency values were observed by Ballesteros et al. [6] for the encapsulation of coffee extracts by freeze-drying, obtaining 62% and 73% efficiency for polyphenols and flavonoids, respectively. On the other hand, Ramírez, Giraldo and Orrego [43] reported that the highest content of phenolic compounds in fruit-extract encapsulates was reached when the compounds were subjected to freeze-drying and using maltodextrin as wall material.

Freeze-drying is an effective technique for the microencapsulation of polyphenolic compounds; this efficiency can be attributed to the changes in morphology caused by the drying process [6,43]. The broken glass shape creates a lower surface area to volume ratio compared with the microspheres obtained by the spray-drying process, since the smaller spheres have a larger contact area for the same amount of material, causing the degradation of the polyphenolic compounds on the surface [6].

### 2.4. TPC, TFC and In Vitro Relative Bio-Accessibility (RB)

LE presented the highest content of TPC and TFC when compared with microencapsulated lemon extracts (Figure 1 and Table 2). However, the initial weighing of the microcapsules to determine TPC and TFC includes the wall material (maltodextrin), which is removed prior to testing. For this reason, the microencapsulated lemon extracts presented lower TPC and TFC than LE. The same behavior for phenolic compounds was reported by Bernardes et al. [44], who indicated that the microcapsules of jussara extract (*Euterpe edulis* Martius) presented a lower total phenolic content than the extract.

Table 2 shows significant differences for different extraction methods at the same maltodextrin concentration (rows) are indicated by different lowercase letters. TPC and TFC were significantly lower (*p* < 0.05) for lemon extract obtained by CE and microencapsulated with the same maltodextrin concentration, resulting in microcapsules with lower relative bio-accessibility when compared with the microcapsules with lemon extract obtained by UAE and HHPE. The significant differences for different maltodextrin concentration using the same extraction method (columns) are indicated by different uppercase letters within a Table 2. TPC and TFC were increased with an increasing the maltodextrin concentration (M10, M20 and M30) and consequently positively affected the bio-accessibility of bioactive compounds.

The in vitro RB of TPC and TFC in microencapsulated lemon extracts with M10, M20 and M30 are shown in Table 2. After in vitro digestion, the RB of TPC and TFC decreased for all samples. However, the TPC exhibits RB values higher than 50%, whereas the RB of TFC did not exceed 30%. The lemon extracts microencapsulated with M30 show the highest values of TPC and TFC regardless of the extraction method. The highest content of maltodextrin (M30) required a longer time to solubilize the lemon extract microcapsules based on the results reviewed in the Section 2.2.2. Therefore, this property decreases the time of exposure of the compounds to the conditions of gastrointestinal digestion. Hence, the use of microencapsulation is a good tool to protect phenolic compounds from gastrointestinal environmental conditions and increase their bio-accessibility, and, consequently, bioavailability.

## 3. Materials and Methods

### 3.1. Raw Material and Sample Preparation

Lemons (*Citrus limon*, Génova variety) were purchased from local markets in La Serena, Chile. Samples were homogeneously selected based on harvest date (March 2018), color, size and freshness (without mechanical or microbiological damage), as measured by visual analysis. Next, each lemon was divided into pulp, peel and seeds. The pulp was mixed in a rotor-stator homogenizer (Ultraturrax, T25, IKA, Staufen, Germany) at full power for 3 × 15 s. The lemon pulp was characterized by values for moisture of 91.4 g, crude protein of 0.8 g, total lipids of 0.1 g, crude fiber of 1.3 g, crude ash of 0.4 g and carbohydrates (by difference) of 6.0 g per 100 g FW. The peel and seeds were discarded.

### 3.2. Lemon Extract (LE) Preparation

The lemon pulp was homogenized for 30 s at a 1:2 (*w*/*v*) ratio in aqueous methanol (80%). LEs were obtained by three methods [45]: (a) conventional extraction (CE), where samples were extracted by orbital shaking (200 rpm) at room temperature (RT; 20 ± 2 °C) for 120 min; (b) ultrasound-assisted extraction (UAE), where the extraction was performed during 15 min in an ultrasound bath (Branson 2510 E-MT, 42 kHz, 130 W; Danbury, CT, USA) at RT; and (c) high-hydrostatic-pressure extraction (HHPE), where samples were packaged in high-density polyethylene bags and placed in a 2 L cylindrical loading container at RT and pressurized at 500 MPa for 15 (Avure Technologies, Kent, Washington, CA, USA) min with 1 min pulses. Finally, all samples were centrifuged (15 min at 20 °C, 6000 *g*) and supernatants were recovered and evaporated to dryness using a rotary evaporator (Büchi RE12, Flawil, Switzerland) under reduced pressure at 40 °C.

### 3.3. Microencapsulation of LE

Mixtures at a 1:2 (*w*/*w*) ratio between lemon extract and maltodextrin solution were prepared. Three maltodextrin concentrations were employed: 10%, 20% and 30% (*w*/*v*), identified as M10, M20 and M30, respectively, according to the results obtained by previous studies (data not shown). All samples were stirred gently with a magnetic stirrer during 30 min to dissolve solid particles prior to freeze-drying. Subsequently, the freeze-drying process was performed over 20 h (Benchtop 3 L, New York, NY, USA; −52 °C and 200 mbar). Dried samples were ground (IKA^®^ A-11, Wilmington, DE, USA). Then, the microcapsules were collected, packed in Falcon tubes and stored in the dark in a desiccator.

### 3.4. Microcapsule Analysis

#### 3.4.1. Moisture Content and Water Activity

The moisture content was determined by AOAC method no. 934.06 [46]. Water activity (a_w_) was determined at 25 °C by using a water activity analyzer (4TE, AquaLab, Pullman, WA, USA). All measurements were made in triplicate.

#### 3.4.2. Dissolution Time

Two grams of microcapsules were mixed in 50 mL of distilled water with a magnetic stirrer (OS-100, HiLab, Indonesia) at 892 rpm and a stir bar measuring 2 mm × 7 mm [31]. The time required for the complete dissolution of microcapsules in water (observed clarity of the solution) was evaluated.

#### 3.4.3. Hygroscopicity Assay

One gram of microcapsules was placed in an airtight glass container with NaCl saturated solution (75.3%) at RT. After 7 days, the microcapsules were weighed and their hygroscopicity was expressed as grams of adsorbed moisture per 100 g of dry solids (g/100 g) [31].

#### 3.4.4. Morphology and Particle Size

The morphology and particle size of the microcapsules were observed with a scanning electron microscope (Hitachi SU3500, Tokyo, Japan) at an accelerating voltage of 25 kV. Prior to using the scanning electron microscope (SEM), samples were placed on a stub and coated with platinum/palladium using a vacuum-sputtering coater (With Fine Coat Ion Sputter JFC-1100, Tokyo, Japan). Images were processed to estimate the particle sizes using the ZEISS Zen, version 3.3 (blue edition) (Software for image acquisition, processing, and analysis; Carl Zeiss: Jena, Germany).

### 3.5. Determination of Total Polyphenolic and Flavonoid Content

The coating material structure (M10, M20 and M30) of the microcapsules was completely removed as reported by Nunes et al. [31]. Two hundred milligrams of microcapsules were vortex-mixed for 60 s in 1 mL of ethanol: acetic acid: water (50:8:42 *v*/*v*/*v*) then ultrasonicated (Branson 2510 E-MT, 42 kHz, 130 W; Danbury, CT, USA) for 20 min. The supernatant was centrifuged at 6000× *g* for 5 min and filtered (Whatman No. 1, Springfield Mill, Maidstone Kent, UK). The solvent was evaporated in a rotary evaporator (Büchi RE12, Flawil, Switzerland) under reduced pressure at 40 °C. The dry extract was redissolved in aqueous methanol (80%) to a final volume of 10 mL.

Total polyphenolic contents (TPC) of LE (lemon extract) and LEM (microencapsulated lemon extract) were determined using Folin–Ciocalteu’s method [47]. Gallic acid was used to construct a standard curve (50–1000 mg/mL) and total polyphenolic content was expressed as mg gallic acid equivalents (GAE) g^−1^. All measurements were performed in triplicate.

Total flavonoid contents (TFC) of LE and LEM were determined by aluminum chloride colorimetric assay [48]. A standard curve was plotted using quercetin standard (20–100 μg mL^−1^) and total flavonoid content was expressed as mg quercetin equivalents (QE) g^−1^. All measurements were performed in triplicate.

### 3.6. In Vitro Digestion and Relative Bio-Accessibility (RB)

In vitro digestion simulates the different phases of digestion (oral, gastric and intestinal, small intestine) according to the protocol proposed by Minekus et al. [49]. To quantify bio-accessibility, the recovery of total bioactive compounds was estimated in the samples before and after in vitro digestion. RB percentages were calculated as follows [50]:$$\mathrm{RB}\left(\%\right)=\frac{\mathrm{CD}}{\mathrm{CI}}\times 100$$
where CD is the concentration of bioactive compounds after digestion (bio-accessible fraction) and CI is the initial concentration of bioactive compounds before digestion.

### 3.7. Microencapsulation Efficiency (ME)

The MEs for TPC and TFC were calculated using the following equations [15]:$${\mathrm{ME}}_{\mathrm{TPC}}\left(\%\right)=\frac{{\mathrm{TPC}}_{M}}{{\mathrm{TPC}}_{E}}\times 100$$
$${\mathrm{ME}}_{\mathrm{TFC}}\left(\%\right)=\frac{{\mathrm{TFC}}_{M}}{{\mathrm{TFC}}_{E}}\times 100$$
where TPC_M_ and TFC_M_ are the amounts of TPC and TFC, respectively, in the microcapsules, whereas TPC_E_ and TFC_E_ are the amounts of TPC and TFC, respectively, in the extracts before microencapsulation.

### 3.8. Statistical Analysis

Data analysis was carried out using Statgraphics Plus^®^ 5.1 software. Analysis of variance (ANOVA) was used to determine significant differences (*p* < 0.05) between the microcapsules and controls (extract). Differences between the means were detected using Fisher’s test. Additionally, the multiple range test (MRT) was used to find homogeneous groups within each of the analyzed parameters.

## 4. Conclusions

This research shows that ultrasound and high hydrostatic pressure increase the extractability of the phenolic and flavonoid content from lemon extract and, consequently, increase the yield of the extract’s microencapsulation process. For all microcapsules, the addition of maltodextrin decreased the moisture content, water activity and hygroscopicity, whereas the time required to solubilize in water increased. Additionally, the microcapsules showed irregular shapes and different particle sizes, which is characteristic of microcapsules produced by the freeze-drying process. The LEs obtained by UAE and HHPE presented an increase in TPC and TFC, and therefore higher microencapsulation efficiencies were observed when compared with the extract obtained by CE. The freeze-drying process efficiently microencapsulated the bioactive compounds obtained from lemon extract, with high TPC and TFC retention (92.3 and 83.1%, respectively) for M30. However, there was no significant difference in the microencapsulation efficiency of the lemon extract obtained by the UAE and HHPE methods. After in vitro digestion, the relative bio-accessibility of TPC and TFC decreased for all samples. These results highlight the importance of the microencapsulation of lemon extract using maltodextrin because it improves preservation of the phenolic and flavonoid compounds during the freeze-drying process before and after in vitro digestion.

## Figures and Tables

**Figure 1 molecules-27-04166-f001:**
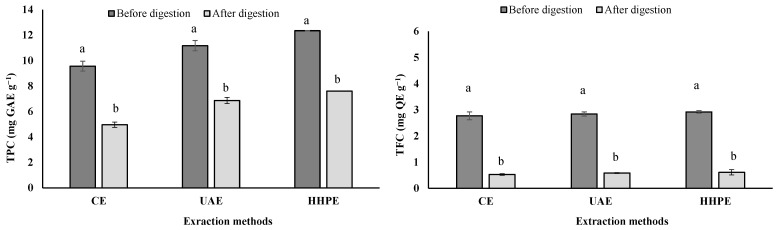
Total phenolic content (TPC) and flavonoid content (TFC) of lemon extract (LE) obtained by CE (conventional extraction), UAE (ultrasound-assisted extraction) and HHPE (high-hydrostatic-pressure extraction) and its content after in vitro digestion. TPC is expressed as milligrams of gallic acid equivalent per gram of sample. TFC is expressed as milligrams of quercetin equivalent per gram of sample. Different lowercase letters on the bars indicate significant differences (*p* < 0.05) between the same group.

**Figure 2 molecules-27-04166-f002:**
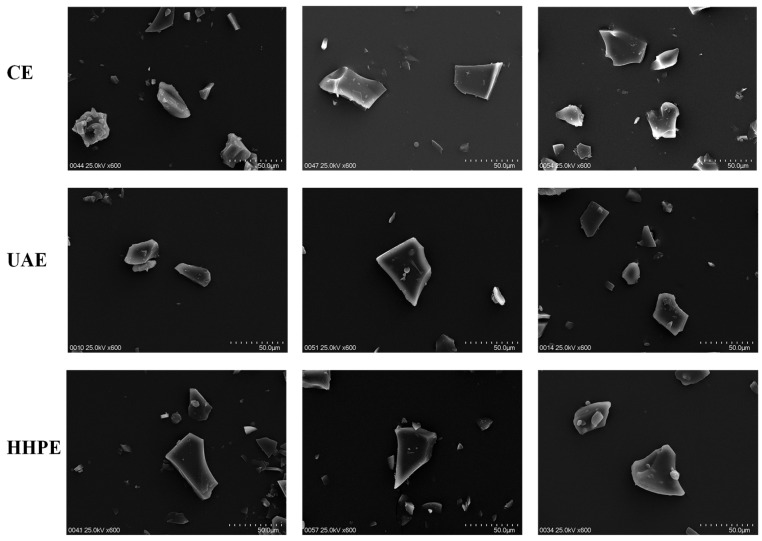
Micrographs by SEM of microcapsules with maltodextrin at 10% (M10), 20% (M20) and 30% (M30) from lemon extract by CE (conventional extraction), UAE (ultrasound-assisted extraction) and HHPE (high-hydrostatic-pressure extraction).

**Table 1 molecules-27-04166-t001:** Mean and standard deviation (brackets) of water activity, moisture content (g 100 g^−1^), solubility in water (s), hygroscopicity (g 100 g^−1^) of microcapsules with maltodextrin at 10% (M10), 20% (M20) and 30% (M30) from lemon extract by CE (conventional extraction), UAE (ultrasound-assisted extraction) and HHPE (high-hydrostatic-pressure extraction).

		Properties				
Extraction Method	Coating	Water Activity (a_w_)	Moisture Content (g 100 g^−1^)	Solubility in Water (s)	Hygroscopicity (g 100 g^−1^)	Particle Size (µm)
CE	M10	0.299 (0.002) ^a^	3.72 (0.15) ^a^	135.0 (1.4) ^a^	6.30 (0.34) ^a^	32.17 (4.63) ^a^
	M20	0.296 (0.002) ^a^	3.71 (0.15) ^a^	225.2 (2.9) ^b^	6.19 (0.39) ^a^	41.78 (4.88) ^b^
	M30	0.283 (0.002) ^b^	3.65 (0.14) ^a^	232.5 (9.1) ^b^	6.14 (0.55) ^a^	43.58 (6.33) ^b^
UAE	M10	0.298 (0.001) ^a^	3.66 (0.10) ^a^	137.1 (5.0) ^a^	6.28 (0.24) ^a^	35.85 (3.20) ^a^
	M20	0.297 (0.002) ^a^	3.60 (0.20) ^a^	229.5 (3.3) ^b^	6.26 (0.57) ^a^	44.00 (4.95) ^b^
	M30	0.284 (0.003) ^b^	3.57 (0.30) ^a^	233.4 (7.8) ^b^	6.22 (0.59) ^a^	48.29 (8.20) ^b^
HHPE	M10	0.299 (0.002) ^a^	3.72 (0.08) ^a^	138.0 (2.3) ^a^	6.32 (0.24) ^a^	34.56 (3.44) ^a^
	M20	0.295 (0.003) ^a^	3.60 (0.06) ^a^	227.4 (2.9) ^b^	6.18 (0.14) ^a^	43.42 (5.20) ^b^
	M30	0.282 (0.002) ^b^	3.59 (0.10) ^a^	235.1 (6.0) ^b^	6.14 (0.33) ^a^	47.77 (9.40) ^b^

Different superscript letters in the columns indicate significant differences (*p* < 0.05) between the microcapsules.

**Table 2 molecules-27-04166-t002:** Mean and standard deviation (brackets) of total phenolic content (TPC), total flavonoid content (TFC) and their relative bio-accessibility (RB) of lemon extract (LE) obtained by CE (conventional extraction), UAE (ultrasound-assisted extraction) and HHPE (high-hydrostatic-pressure extraction) and microencapsulated with maltodextrin at 10% (M10), 20% (M20) and 30% (M30).

		Extraction Method					
		CE		UAE		HHPE	
Assay	Sample	Value	RB (%)	Value	RB (%)	Value	RB (%)
TPC (mg GAE g^−1^)	M10	7.23 (0.22) ^a.A^	65.0	8.78 (0.04) ^b.A^	69.2	9.55 (0.09) ^c.A^	70.0
	M20	8.55 (0.30) ^a.B^	71.2	10.23 (0.30) ^b.B^	74.5	11.05 (0.20) ^c.B^	74.9
	M30	8.87(0.15) ^a.B^	73.3	10.61 (0.50) ^b.AB^	76.1	11.68 (0.20) ^c.C^	77.2
TFC (mg QE g^−1^)	M10	1.99 (0.01) ^a.A^	20.5	2.05 (0.01) ^b.A^	22.0	2.27 (0.01) ^c.A^	21.3
	M20	2.29 (0.02) ^a.B^	22.1	2.26 (0.04) ^b.B^	25.1	2.39 (0.02) ^c.B^	25.7
	M30	2.35 (0.01) ^a.C^	23.4	2.32 (0.01) ^b.C^	25.4	2.48 (0.01) ^c.C^	25.2

Different lowercase and uppercase letters in columns indicate significant differences between the mean values (*p* < 0.05). TPC is expressed as milligrams of gallic acid equivalent per gram of sample. TFC is expressed as milligrams of quercetin equivalent per gram of sample.

## Data Availability

Not applicable.

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
