# Peer review of "Effect of Extraction Methods and In Vitro Bio-Accessibility of Microencapsulated Lemon Extract"

_molecules, 2022, doi:10.3390/molecules27134166_

Round 1

Reviewer 1 Report

The authors have presented an interesting study. Overall, the conclusion are supported by the results shown. However, there are several points that the authors need to address in a revised manuscript. These points are listed below.

1. Page 2, line 57: (CE), (UAE), and (HHPE) should be inserted as these abbreviations are used after the point in the manuscript.

2. Figure 1: The y-axis label should be GAE to match that mentioned in the Methods section. CE, UAE, HHPE should be defined in the figure caption.

3. Page 3, line 104: aw should be defined here.

4. Table 2: Some additional text describing this table should be provided to discuss the statistics and which pairs are being compared. It is not clear to a more general reader.

Author Response

Reviewer 1: Comments/ Answer

We thank the Reviewer for their interest in our work and for helpful comments that will greatly improve the manuscript and we have tried to do our best to respond to the points raised.

As indicated below, we have checked all the comments provided by the Reviewer and have made necessary changes accordingly to their indications:

Comment 1

Page 2, line 57: (CE), (UAE), and (HHPE) should be inserted as these abbreviations are used after the point in the manuscript.

Answer

Line 57- The abbreviations were added in introduction section.

Comment 2

Figure 1: The y-axis label should be GAE to match that mentioned in the Methods section. CE, UAE, HHPE should be defined in the figure caption.

Answer

Figure 1- The y-axis label was modified. CE, UAE, HHPE were defined in the figure caption.

Comment 3

Page 3, line 104: aw should be defined here.

Answer

Line 104- The aw definition was added “Water activity corresponds the amount of free water in a food system, this available water is a factor that contributes to food deterioration because it can be used by unwanted microorganisms [19].”

Comment 4

Table 2: Some additional text describing this table should be provided to discuss the statistics and which pairs are being compared. It is not clear to a more general reader.

Answer

Table 2- The paragraph was added in results and discussion: “Table 2 shows significant differences for different extraction methods at the same maltodextrin concentration (rows) are indicated by different lowercase letters. TPC and TFC were significantly lower (p<0.05) for lemon extract obtained by CE and microencapsulated with the same maltodextrin concentration, resulting in microcapsules with lower relative bio-accessibility when compared to the microcapsules with lemon extract obtained by UAE and HHPE. While the significant differences for different maltodextrin concentration in the same extraction method (columns) are indicated different uppercase letters within a Table 2. TPC and TFC were increased with an increasing the maltodextrin concentration (M10, M20 and M30) and consequently positively affected the bio-accessibility of bioactive compounds.”

Reviewer 2 Report

In this manuscript, authors reported the effect of different extraction methods and in vitro bio-accessibility of microencapsulated lemon extract. This is an interesting and well-performed study. I have a few minor comments (below), but overall, I think it is eminently worthy of publication.

1. In section “3.2. Lemon extract (LE) preparation”, has the preparation method and its conditions and parameters provided by the authors been validated? Are they optimal?

2. The section “3.3. Microencapsulation of LE” has the same problem.

3. Room temperature (RT) should be given a temperature range, because in different countries, regions, and different seasons, its meaning may be different.

4. There is a problem of poor representativeness in sample particle size measurement by SEM. Authors should give a detailed description to explain how to solve this problem.

5. What are the contents of gallic acid and quercetin in LE and LEM, respectively? Is it reasonable to use them as standards to determine the total polyphenolic acid and flavonoid content?

Author Response

Reviewer 2: Comments/ Answer

We thank the Reviewer for their interest in our work and for helpful comments that will greatly improve the manuscript and we have tried to do our best to respond to the points raised.

As indicated below, we have checked all the comments provided by the Reviewer and have made necessary changes accordingly to their indications:

Comment 1

  1. In section “3.2. Lemon extract (LE) preparation”, has the preparation method and its conditions and parameters provided by the authors been validated? Are they optimal?

Answer

The extraction protocol was previously published. The reference was included in the manuscript (Line 261).

Comment 2

  1. The section “3.3. Microencapsulation of LE” has the same problem.

Answer

Previous lab tests were carried out for microencapsulation procedure of lemon extract. The Lab tests were based on previous research that had reported the use of maltodextrin as a wall material in plant matrices. Therefore, the sentence was added in the manuscript: “…according to the results obtained by previous studies (data not shown).”

Comment 3

  1. Room temperature (RT) should be given a temperature range, because in different countries, regions, and different seasons, its meaning may be different.

Answer

The room temperature was added in manuscript: “… at room temperature (RT; 20 ± 2 °C)…”

Comment 4

  1. There is a problem of poor representativeness in sample particle size measurement by SEM. Authors should give a detailed description to explain how to solve this problem.

Answer

*In this study we used only SEM analysis to verify the size of the microcapsules after freeze-drying, in a first instance, the SEM software (Stitch software) was used to measure the particle size; accordingly, to how manuscript has been evaluating was performed by Nunes et al. (2015).  However, we got better at particle size analysis using the image analysis software, ZEISS Zen, version 3.3 (blue edition) (Carl Zeiss: Jena, Germany).  This analysis was added in manuscript (Methodology, Table 1, results).

**The information was added in methodology: “Images were processed to estimate the particle sizes using the ZEISS Zen, version 3.3 (blue edition) (Software for image acquisition, processing, and analysis; Carl Zeiss: Jena, Ger-many).”

Comment 5

  1. What are the contents of gallic acid and quercetin in LE and LEM, respectively? Is it reasonable to use them as standards to determine the total polyphenolic acid and flavonoid content?

Answer

The criteria considered in the choice of the standard were the availability, stability,  price,  solubility in the solvent used and the presence in the sample. According to the literature, both polyphenols have been detected in lemon pulp of different varieties. In addition, the results were expressed in an unit commonly used,  to compare with literature data [1–3].

References

  1. Makni, M.; Jemai, R.; Kriaa, W.; Chtourou, Y.; Fetoui, H. Citrus limon from Tunisia: Phytochemical and Physicochemical Properties and Biological Activities. Biomed Res. Int. 2018, 2018, doi:10.1155/2018/6251546.
  2. Brito, A.; Ramirez, J.E.; Areche, C.; Sepúlveda, B.; Simirgiotis, M.J. HPLC-UV-MS profiles of phenolic compounds and antioxidant activity of fruits from three citrus species consumed in Northern Chile. Molecules 2014, 19, 17400–17421, doi:10.3390/molecules191117400.
  3. Czech, A.; Malik, A.; Sosnowska, B.; Domaradzki, P. Bioactive Substances, Heavy Metals, and Antioxidant Activity in Whole Fruit, Peel, and Pulp of Citrus Fruits. Int. J. Food Sci. 2021, 2021, doi:10.1155/2021/6662259.

Reviewer 3 Report

The manuscript molecules-1751679 brings the use of emerging technologies to improve the characteristics of lemon extracts, also evaluating the effect of such methods on the properties of microcapsules. The authors performed good work, but the design of the study flaws and missed some details on the M&M, which may compromise the article, as indicated below.

Firstly, the authors did not indicate how the conditions for the extractions were chosen. Previous studies? Lab tests? Were the conditions previously optimized? Experimental designs?

Secondly, the study used only SEM analysis to verify the size of the microcapsules after freeze-drying. However, a particle size analyzer such as Mastersizer must be used to evaluate the particle size distribution before the drying processes to confirm the micro-scale. It is also indicated to use a Zetasizer to verify the stability of the encapsulated mixture.

L14-21 Abstract – Please provide a short sentence on the background to show the readers the importance of using the chosen raw material and the emerging techniques to enhance the properties of the obtained products. Clearly indicate “orbital shaking” as your conventional extraction, as many other techniques may be “conventional.” Also, finish the abstract suggesting where those microcapsules could be applied.

L23 Keywords – I suggest replacing “fruit” with“citrus,” as fruit is too general. Also, include “ultrasound” and “high-hydrostatic pressure”

L26-33 Introduction – Please include more recent data on lemon production/consumption, which is available from USDA (https://apps.fas.usda.gov/psdonline/circulars/citrus.pdf) and Statista (https://www.statista.com/statistics/577445/world-lemon-and-lime-production/)

L34-45 – The authors provided a discussion on the encapsulation method (freeze-drying), but no mention of the extraction methods of the bioactive compounds (which is in the title of the article) has been made. There is plenty of literature on the topic. The authors must include sentences on the methods used and indicate their impact on the quality and characteristics, as well as the stability of the extracted compounds, also impacting the quality of the obtained microcapsules.

L74 – Please indicate “in vitro” and all the other Latin names in italics.

L84-86 – Please use the author’s quotations according to the Journals guideline (e.g., Ovandro-Martínez et al. [13])

L91-92 – Figure 1 – Please indicate in the caption what the acronyms stand for (TPC, TFC, AG, QE, etc.), as figures and tables must be self-explaining

L97 – Water activity abbreviation must be indicated with the subscript letter w.

L113-127 – A discussion on the non-difference in the solubility among M20 and M30 must be included, as the reduced use of raw materials to obtain the desired effect can be a positive result.

L142-144 – The authors did not indicate in the table what is the statistical parameter in parenthesis (standard deviation, standard error?). The extraction methods (CE, UAE, and HHPE) must also be written in full in the title, as the tables must be self-explaining (as in Table 2). Also, change the order between hygroscopicity and solubility to follow the same order of the discussion.

L199-200 – Please indicate scientific names appropriately according to the Binominal Nomenclature rules, showing only genus and species in italics.

L259 – Correct degree sign and double-check the manuscript for the correct use of SI units (for example, italics in “g” when it stands for “gravity” to differentiate it from “gram”)

L277 – Indicate how the authors measured the particle size in the SEM section (software used, for example).

L323 – Rewrite the beginning of the conclusion as the broken sentence used is quite odd.

Author Response

Reviewer 3: Comments/ Answer

We thank the Reviewer for their interest in our work and for helpful comments that will greatly improve the manuscript and we have tried to do our best to respond to the points raised.

As indicated below, we have checked all the comments provided by the Reviewer and have made necessary changes accordingly to their indications:

Comment 1

The authors did not indicate how the conditions for the extractions were chosen. Previous studies? Lab tests? Were the conditions previously optimized? Experimental designs?

Answer

The extraction protocol was a previously published study. The reference was included in the manuscript (Line 261).

Comment 2

The study used only SEM analysis to verify the size of the microcapsules after freeze-drying. However, a particle size analyzer such as Mastersizer must be used to evaluate the particle size distribution before the drying processes to confirm the micro-scale. It is also indicated to use a Zetasizer to verify the stability of the encapsulated mixture.

Answer

In this study we used only SEM analysis to verify the size of the microcapsules after freeze-drying, in a first instance, the SEM software (Stitch software) was used to measure the particle size; accordingly, to how manuscript has been evaluating was performed by Nunes et al. (2015).  However, we got better at particle size analysis using the image analysis software, ZEISS Zen, version 3.3 (blue edition) (Carl Zeiss: Jena, Germany).  This analysis was added in manuscript (Methodology, Table 1, results).

We would consider implemented particle size analysis in further studies, since we currently have no available instrument for such test.

Nunes, G.L.; Boaventura, B.C.B.; Pinto, S.S.; Verruck, S.; Murakami, F.S.; Prudêncio, E.S.; De Mello Castanho Amboni, R.D. Microencapsulation of freeze concentrated Ilex paraguariensis extract by spray drying. J. Food Eng. 2015, 151, 60–68, doi:10.1016/j.jfoodeng.2014.10.031.

Comment 3

*L14-21 Abstract – Please provide a short sentence on the background to show the readers the importance of using the chosen raw material and the emerging techniques to enhance the properties of the obtained products.

**Clearly indicate “orbital shaking” as your conventional extraction, as many other techniques may be “conventional.”

 ***Also, finish the abstract suggesting where those microcapsules could be applied.

Answer

* L14-L21. The text was added in abstract: “The extraction of bioactive compounds from fruits, such as lemon, has gained relevance because these compounds have beneficial properties for health, such as antioxidant and anticancer properties, however, the extraction method can significantly affect these properties. High hydrostatic pressure and ultrasound as emerging extraction methods constitute an alternative to conventional extraction, improving extractability and obtaining extracts rich in bioactive compounds. Therefore, ...”

**The orbital shaking was added in abstract.

*** This text was added: “In addition, this freeze-dried product can be utilized as functional ingredient for food or supplement formulations.”

Comment 4

L23 Keywords – I suggest replacing “fruit” with “citrus,” as fruit is too general. Also, include “ultrasound” and “high-hydrostatic pressure”

Answer

Thanks for the suggestion, the keywords were included in the manuscript by: Keywords: “Citrus; Microcapsule; In-vitro digestion; Emerging Technologies; ultrasound; high-hydrostatic pressure.”

Comment 5

L26-33 Introduction – Please include more recent data on lemon production/consumption, which is available from USDA (https://apps.fas.usda.gov/psdonline/circulars/citrus.pdf) and Statista (https://www.statista.com/statistics/577445/world-lemon-and-lime-production/)

Answer

Thanks for information. L23-33 - The data was added in the introduction: “In 2021, the Food and Agriculture Organization's corporate statistics database reported that the world production of lemons and limes was approximately 21 million tons [1]. Chile is among the ten citrus producers in South America, exporting 366,378 tons of citrus in 2020, of which 26% corresponds to lemons [1,2] . The main varieties of lemon grown in the Chile are Genova, Eureka, Fino, and Messina [3].

References

  1. 1. FAO (Food and Agriculture Organization of the United Nations (FAO). Citrus Fruit Statistical Compendium 2020. Bull. 2021, 1–40, doi:10.5860/choice.36-2167.
  2. Fruits From Chile Citrus Available online: https://fruitsfromchile.com/fruit/citrus/ (accessed on Jun 1, 2022).
  3. Comité de Cítricos LIMONES Available online: https://www.comitedecitricos.cl/productos/limones (accessed on Jun 1, 2022).

Comment 6

L34-45 – The authors provided a discussion on the encapsulation method (freeze-drying), but no mention of the extraction methods of the bioactive compounds (which is in the title of the article) has been made. There is plenty of literature on the topic. The authors must include sentences on the methods used and indicate their impact on the quality and characteristics, as well as the stability of the extracted compounds, also impacting the quality of the obtained microcapsules.

Answer

The text was added in the introduction: “The extraction method can significantly affect the bioactive compounds and their properties. High hydrostatic pressure (HHPE) and ultrasound-assisted (UAE) extractions have been used to maximize the recovery of bioactive compounds present in the fruit and their by-products [7]. Both extraction techniques seek to increase the solubility of the bio-active compounds while improving mass transfer [8]. Besides, EAU and HHPE compared to conventional extraction (heat or/and agitation) have been reported to provide higher ex-traction efficiency in less time, with less solvent and without high temperatures. HHPE and UAE have been successfully applied in the extraction of pectin and polyphenols from tomato peel waste [7], lycopene and flavonoids from tomato pulp [9], polyphenols from lemon, lime, tangerine, and sweet orange peel [10,11], polyphenols from sweet orange [12] and anthocyanins from jambolan fruit [13].”

References

  1. Ninčević Grassino, A.; Ostojić, J.; Miletić, V.; Djaković, S.; Bosiljkov, T.; Zorić, Z.; Ježek, D.; Rimac Brnčić, S.; Brnčić, M. Application of high hydrostatic pressure and ultrasound-assisted extractions as a novel approach for pectin and polyphenols recovery from tomato peel waste. Innov. Food Sci. Emerg. Technol. 2020, 64, doi:10.1016/j.ifset.2020.102424.
  2. Giacometti, J.; Bursać Kovačević, D.; Putnik, P.; Gabrić, D.; Bilušić, T.; Krešić, G.; Stulić, V.; Barba, F.J.; Chemat, F.; Barbosa-Cánovas, G.; et al. Extraction of bioactive compounds and essential oils from mediterranean herbs by conventional and green innovative techniques: A review. Food Res. Int. 2018, 113, 245–262, doi:10.1016/j.foodres.2018.06.036.
  3. Briones-Labarca, V.; Giovagnoli-Vicuña, C.; Cañas-Sarazúa, R. Optimization of extraction yield, flavonoids and lycopene from tomato pulp by high hydrostatic pressure-assisted extraction. Food Chem. 2019, 278, doi:10.1016/j.foodchem.2018.11.106.
  4. Casquete, R.; Castro, S.M.; Villalobos, M.C.; Serradilla, M.J.; Queirós, R.P.; Saraiva, J.A.; Códoba, M.G.; Teixeira, P. High Pressure Research : An High pressure extraction of phenolic compounds from citrus peels †. High Press. Res. 2014, 34, 447–451, doi:10.1080/08957959.2014.986474.
  5. Casquete, R.; Castro, S.M.; Martín, A.; Ruiz-Moyano, S.; Saraiva, J.A.; Córdoba, M.G.; Teixeira, P. Evaluation of the effect of high pressure on total phenolic content, antioxidant and antimicrobial activity of citrus peels. Innov. Food Sci. Emerg. Technol. 2015, 31, 37–44, doi:10.1016/J.IFSET.2015.07.005.
  6. Nishad, J.; Saha, S.; Kaur, C. Enzyme- and ultrasound-assisted extractions of polyphenols from Citrus sinensis (cv. Malta) peel: A comparative study. J. Food Process. Preserv. 2019, 43, e14046, doi:10.1111/JFPP.14046.
  7. Sabino, L.B. de S.; Filho, E.G.A.; Fernandes, F.A.N.; de Brito, E.S.; Júnior, I.J. da S. Optimization of pressurized liquid extraction and ultrasound methods for recovery of anthocyanins present in jambolan fruit (Syzygium cumini L.). Food Bioprod. Process. 2021, 127, 77–89, doi:10.1016/J.FBP.2021.02.012.

Comment 7

L74 – Please indicate “in vitro” and all the other Latin names in italics.

Answer

The text was changed in the manuscript.

Comment 8

L84-86 – Please use the author’s quotations according to the Journals guideline (e.g., Ovandro-Martínez et al. [13])

Answer

Thanks, the reference was corrected according to the Journals guideline.

Comment 9

L91-92 – Figure 1 – Please indicate in the caption what the acronyms stand for (TPC, TFC, AG, QE, etc.), as figures and tables must be self-explaining

Answer

Thanks, the captions were corrected in figures and tables.

Comment 10

L97 – Water activity abbreviation must be indicated with the subscript letter w.

Answer

Water activity abbreviation was corrected in L97 of manuscript.

Comment 11

L113-127 – A discussion on the non-difference in the solubility among M20 and M30 must be included, as the reduced use of raw materials to obtain the desired effect can be a positive result.

Answer

L113-L127. This text was added: “No significant differences (p>0.05) were observed between the M20 and M30. This result shows that the addition of a smaller amount of wall material (M20) can reduce the raw material cost without changing the solubility of the microcapsules.”

Comment 12

L142-144 – The authors did not indicate in the table what is the statistical parameter in parenthesis (standard deviation, standard error?). The extraction methods (CE, UAE, and HHPE) must also be written in full in the title, as the tables must be self-explaining (as in Table 2). Also, change the order between hygroscopicity and solubility to follow the same order of the discussion.

Answer

L142-144. The information in Table of statistical parameter and extraction method names were added in the title.

The order between hygroscopicity and solubility was changed to follow the same order of the discussion.

Comment 13

L199-200 – Please indicate scientific names appropriately according to the Binominal Nomenclature rules, showing only genus and species in italics.

Answer

Scientific name was corrected in L199-200 of the manuscript.

Comment 14

L259 – Correct degree sign and double-check the manuscript for the correct use of SI units (for example, italics in “g” when it stands for “gravity” to differentiate it from “gram”)

Answer  

The degree was corrected in L59. In addition, the manuscript was checked.

Comment 15

L277 – Indicate how the authors measured the particle size in the SEM section (software used, for example).

Answer

The information was added in manuscript: “Images were processed to estimate the particle sizes using the ZEISS Zen, version 3.3 (blue edition) (Software for image acquisition, processing, and analysis; Carl Zeiss: Jena, Ger-many).”

Comment 16

L323 – Rewrite the beginning of the conclusion as the broken sentence used is quite odd

Answer

In L323. This text was added in conclusion: “This research shows that ultrasound and high hydrostatic pressure increase the extractability of the phenolic and flavonoid content from lemon extract and, consequently, increase the yield of the extract's microencapsulation process. For all microcapsules, …. “

Round 2

Reviewer 1 Report

The authors have adequately addressed previous concerns. The overall quality of the manuscript has been improved.

Author Response

We thank the Reviewer for their interest in our research.

Reviewer 3 Report

L33-38 – Scientific names of plant species must be given at their first appearance.

L50 – Correct EAU to UAE

Table 2 – Indicate TPC and TFC equivalents in the footnote rather than the title.

Author Response

We thank the Reviewer. We have reviewed the comments and made the changes according to the indications:

Comment 1

L33-38 – Scientific names of plant species must be given at their first appearance.

Answer

Line 33-38 - Scientific names of plant species were added.

Comment 2

L50 – Correct EAU to UAE

Answer

Thanks, the word was corrected (L50).

Comment 3

Table 2 – Indicate TPC and TFC equivalents in the footnote rather than the title

Answer

Table 2 - TPC and TFC equivalents were moved to footnote.